# Crowdsourcing with Difficulty: A Bayesian Rating Model for Heterogeneous Items

## Abstract

In applied statistics and machine learning, the "gold standards" used for training are often biased and almost always noisy. Dawid and Skene's justifiably popular crowdsourcing model adjusts for rater (coder, annotator) sensitivity and specificity, but fails to capture distributional properties of rating data gathered for training, which in turn biases training. In this study, we introduce a general purpose measurement-error model with which we can infer consensus categories by adding item-level effects for difficulty, discriminativeness, and guessability. We further show how to constrain the bimodal posterior of these models to avoid (or if necessary, allow) adversarial raters. We validate our model's goodness of fit with posterior predictive checks, the Bayesian analogue of $\chi^2$ tests, and assess its predictive accuracy using leave-one-out cross-validation. We illustrate our new model with two well-studied data sets, binary rating data for caries in dental X-rays and implication in natural language.

## 1 Introduction

Crowdsourcing is the process of soliciting ratings (labels, annotations, codes) from a group of respondents for a set of items and combining them in a useful way for some task, such as training a neural network classifier or a large language model.

The use of the term "crowdsourcing" is not meant to imply that the raters are untrained—the crowd may be one of oncologists, lawyers, or professors. This note focuses on binary ratings, which arise in two-way classification problems, such as whether a patient has a specific condition, whether an image contains a traffic light, whether one consumer product is better than another, whether an action is ethical, or whether one output of a large language model is more truthful than another. Crowdsourcing models are widely applied, for example to construct gold-standard datasets, estimate prevalence, evaluate coding standards, assess rater ability, and analyze item difficulty.

We evaluate with two representative data sets, one for image recognition and one for natural language processing. The first data set involves dental X-rays being rated as positive or negative for caries, a kind of pre-cavity (Espeland & Handelman, 1989). There are thousands of X-rays, each of which is rated positive or negative for caries (a pre-cavity) by the same five dentists. The dentists showed surprisingly little agreement and consensus, especially in cases where at least one dentist rated the X-ray positive for caries. The dentists also varied dramatically in the number of cases each rated as having caries, indicating rater-specific bias toward positive or negative ratings. The second data set consists of pairs of sentences, which are rated positive if the first entails the second. The second data set was collected through Amazon's Mechanical Turk and involves thousands of sentence pairs and dozens of raters, each of whom rated only a subset of the items (Snow et al., 2008). Classification among dentists and natural language semantics are both challenging classification tasks with subjectively defined boundaries, and thus it is not surprising that the rater agreement level is low. The same kind of "bias" arises in rating data for human values as might be used to align a large language model (Batchelder & Romney, 1988).

These examples highlight a central challenge since human ratings are often noisy, biased, and inconsistent across raters and items. Heuristic approaches such as majority voting ignore this uncertainty and can

misrepresent the underlying signal. Crowdsourcing models address the problem by introducing latent true labels along with rater specific sensitivity and specificity and item level parameters such as difficulty, discriminability, and guessability. In this work we ask how crowdsourcing models can be extended to better capture the heterogeneity of raters and items while remaining identifiable. To answer this question, the main contributions of the paper are the following.

- The primary contribution of this work is the expansion of rating models to account for continuous item variation in the form of difficulty, discriminability, and guessability that generalizes discrete easy/hard distinctions. A space of models along five dimensions is considered and thoroughly evaluated using two medium-sized rating data sets. Only models with item-level effects for difficulty and rater effects distinguishing sensitivity and specificity pass the posterior predictive checks (see Section 6). Posterior predictive checks are the Bayesian equivalent of $\chi^2$ goodness-of-fit tests in regression, which test a "null" of the fitted model against the data to evaluate whether the data could have reasonably been generated by the model (Rubin, 1984; Gelman et al., 1996; Formann, 2003).

- Left unconstrained, the parameters of rating models typically admit two modes: one in which raters are cooperative, and another where prevalence is inverted and raters are adversarial. An adversarial rater is one who consistently provides the wrong rating (i.e., they know the answer but intentionally provide the wrong answer). This paper introduces a constraint that allows explicit control over whether adversarial solutions are allowed in which sensitivity is less than one minus specificity. When sensitivity is equal to one minus specificity, raters are pure noise, or "spam". Passonneau & Carpenter (2014) discuss the classification without providing a solution.

- A final contribution is replicable open-source implementations of these models in Stan (Carpenter et al., 2017), which makes them easy to use in R, Python, or Julia.

## 2 Previous work

Rating models show up in multiple fields, including educational testing, from which the model variants introduced here are derived (Lazarsfeld, 1950; Lord et al., 1968; Rasch, 1960). Rating models are widely used in epidemiology, both for multiple diagnostic testing (Albert & Dodd, 2004) and extracting health status information from patient records (Dawid & Skene, 1979). In sociology, rating models were independently developed for cultural consensus theory (Batchelder & Romney, 1988; Romney et al., 1986). More recently, they have become popular for providing human feedback for classification of images (Raykar et al., 2010; Smyth et al., 1994); human ratings are the basis of massive data sets of millions of images and thousands of classes like ImageNet (Deng et al., 2009). Rating models have long been popular for natural language tasks (Passonneau & Carpenter, 2014; Snow et al., 2008).

The only directly related previous work of which we are aware assumes items are a mixture of "easy" items on which all annotators will agree and "hard" items on which they will struggle (Beigman Klebanov & Beigman, 2009). More recently, crowdsourced ratings of language model output are used as a fine-tuning step that adjusts a foundational large language model like GPT-4, which is trained to complete text, into a chatbot like ChatGPT that is (imperfectly) trained to be helpful, truthful, and harmless (Ouyang et al., 2022; Rafailov et al., 2024).

## 3 Applications of crowdsourcing models

There are several tasks to which crowdsourcing models are applied, and for every one of them a rating model improves performance over heuristic baselines. For example, rating models outperform majority voting for category training and outperform indirect measurements like inter-annotator agreement statistics to measure task difficulty (see, e.g., Artstein & Poesio (2008), Sabou et al. (2014), McHugh (2012)).

**Inferring a gold-standard data set.** The first and foremost application of crowdsourcing is to generate a "gold standard" data set, where a single category (or label) is assigned to each item. In terms of generating representative data, it is best to sample data according to its probability (i.e., follow the generative model)

rather than to choose the "best" rating for each item according to a heuristic such as highest probability. The second section of this paper shows that it is better for downstream accuracy to train with a probabilistic corpus that retains information about rating uncertainty. In cases where it is impractical to use a probabilistic corpus with weighted training, we show why it is far better to sample labels according to their posterior probability in the rating model than to choose the "best" label. In particular, we will demonstrate that majority voting schemes among raters are suboptimal compared to sampling, which is in turn dominated by training with the probabilities (a kind of Rao-Blackwellization). If a classifier for the data is available, an even better approach is to jointly train a classifier and rating model, as shown by Raykar et al. (2010).

**Inferring population prevalence.** The second most common application of crowdsourcing is to understand the probability of positivity among items in the population represented by the crowdsourcing data. This is particularly common in epidemiology, where the probability of positive outcomes is the prevalence of the disease in the (sub)population (Albert & Dodd, 2004). It can also be used to analyze the prevalence of hate speech on a social media site or bias in televised news, the prevalence of volcanoes on Venus (Smyth et al., 1994), or the prevalence of positive reviews for a restaurant.

**Understanding and improving the coding standard.** The third most common application of crowdsourcing is to understand the coding standard, which is the rubric under which rating is carried out. Traditionally, this has been measured through inter-annotator agreement (Artstein & Poesio, 2008). In contrast, rating models provide finer-grained analysis of rater accuracy in terms of sensitivity and specificity, as well as the information gain expected from a rating by a specific rater (Passonneau & Carpenter, 2014).

**Understanding and improving raters.** What is the mean sensitivity and specificity and how does it vary among raters? Are sensitivity and specificity anticorrelated or correlated in the population? This understanding can be fed back to the raters themselves for ongoing training. For example, American baseball umpires have extensive feedback on how they call balls and strikes as measured against a very accurate machine's call, which has led to much higher accuracy and consistency among umpires Flannagan et al. (2024). Understanding rater populations, such as those available through Mechanical Turk or Upwork, is important when managing raters for multiple crowdsourcing tasks. For example, it is straightforward with a rating model to infer the proportion of spammers and the proportion of high quality raters. The number of raters required for high quality joint ratings may also be assessed (Passonneau & Carpenter, 2014).

**Understanding the items.** A fifth task, which has received relatively little attention in the crowdsourcing literature, is to understand the structure of the population of items. For example, which items are difficult to rate and why? Which items simply have too little signal to be consistently rated? Which items lie near the decision boundary and which are far away? Which items have high discrimination and why? A discriminative item is one which cleanly separates high ability from low ability raters in their ability to rate it correctly. Understanding the items is the primary focus of educational test design, where the items are test questions and the raters are students. Test questions are selected for standardized tests like the ACT (American College Testing) based on having high discrimination and a useful range of difficulties (ACT, Inc., 2024).

## 4 A general crowdsourcing model

### 4.1 The rating data

Consider a crowdsourcing problem for which there are $I \in \mathbb{N}$ items to rate and $J \in \mathbb{N}$ raters. Long-form data accomodates the varying number of raters per item and the varying number of items per rater. Let $N \in \mathbb{N}$ be the number of ratings, with $\text{rating}_n \in \{0, 1\}$, each of which has a corresponding $\text{rater}_n \in \{1, \ldots, J\}$ and item being rated $\text{item}_n \in \{1, \ldots, I\}$. As shown in Table 1, each row represents an annotation, with columns indicating indices for the rater and item as well as the rating.

### 4.2 Prior crowdsourcing models

Several annotation models have been proposed to model the data annotation process. The simplest multinomial model (Albert & Dodd, 2004) assumes all annotators are equally reliable and does not account for item-level variation. This is the model that leads to equal-weight voting. Dawid & Skene (1979) introduced annotator-

specific sensitivity and specificity, but did not consider item difficulty or guessing. Beigman & Klebanov (2009) model items as "easy" or "regular", with all annotators agreeing on easy items. These models capture important aspects of annotator behavior, but they do not account for continuous item variation or provide a principled way to handle adversarial raters.

### 4.3   Our crowdsourcing model

We propose a general crowdsourcing model that combines rater specific sensitivity and specificity with item level parameters for difficulty, discrimination, and guessing. To ensure identifiability, we include a constraint that rules out adversarial solutions. The data generating process is as follows.

| rater | item | rating |
|-------|------|--------|
| 1 | 1 | 0 |
| 1 | 1 | 1 |
| 15 | 52 | 1 |
| ⋮ | ⋮ | ⋮ |

Table 1: Rating data table in which each row represents a rating by a rater of an item.

**Generating categories.**   For each item, let $z_i \in \{0, 1\}$ denote its latent category, with 1 indicating a "success" or a "positive" result. Let $\pi \in (0, 1)$ be the prevalence of positives. Categories are conditionally independent given the prevalence, $z_i \sim \mathrm{Bernoulli}(\pi)$. For estimation, we marginalize over $z_i$ to form the likelihood used in inference; details are in section 4.4.

**Generating ratings.**   The rating from rater $j$ for item $i$ is generated conditionally given the category $z_i$ of the item. For positive items ($z_i = 1$), sensitivity (i.e., accuracy on positive items) is used, whereas for negative items ($z_i = 0$), specificity (i.e., accuracy on negative items) is used. Thus every rater $j$ will have a sensitivity and specificity $\alpha_j^{\mathrm{sens}}, \alpha_j^{\mathrm{spec}} \in \mathbb{R}$ on the log odds scale (e.g., $\mathrm{logit}^{-1}(\alpha_j^{\mathrm{sens}})$ is the sensitivity). If the sensitivity is higher than the specificity there will be a bias toward 1 ratings, whereas if the specificity is higher than the sensitivity, there is a bias toward 0 ratings. If the model only has sensitivity and specificity parameters that vary by rater, it reduces to the diagnostic testing model of Dawid & Skene (1979). Fixing $\alpha^{\mathrm{sens}} = \alpha^{\mathrm{spec}}$ introduces an unbiasedness assumption whereby a rater has equal sensitivities and specificities.

The items are parameterized with a difficulty $\beta_i \in \mathbb{R}$ on the log odds scale. This difficulty is subtracted from the sensitivity (if $z_i = 1$) or specificity (if $z_i = 0$) as appropriate to give the raw log odds of a correct rating (i.e., a rating matching the true category $z_i$). Fixing $\beta_i = 0$ introduces the (typically erroneous) assumption that every item is equally difficult to rate.

Each item is further parameterized with a positive-constrained discrimination parameter $\delta_i \in (0, \infty)$. This is multiplied by the raw log odds to give a discrimination-adjusted log odds to give a probability of correctly rating the item. With high discrimination, it is more likely a rater with ability greater than the difficulty will get the correct answer and less likely that a rater with ability less than difficulty will get the correct answer. For educational testing, high discrimination test questions are preferable, but for rating wild type data, low discrimination items are common because of natural variations in the signal (e.g., natural language text or an image). Fixing $\delta_i = 1$ introduces the assumption that the items are equally discriminative.

The final parameter associated with an item is a guessability parameter $\lambda_i \in (0, 1)$, giving the probability that a rater can just "guess" the right answer. The probability that a rater assigns the correct rating will thus be the combination of the probability of guessing correctly and otherwise getting the correct answer in the usual way. Fixing $\lambda_i = 0$ introduces the assumption that the raters never guess an answer. Without a guessing parameter, as difficulty goes to infinity, the probability a rater provides the correct label for an item goes to zero. With guessing, the probability of a correct label is always at least the probability of guessing.

The full, unreduced model follows the item-response theory three-parameter logistic model generalized with sensitivity and specificity (which we denote "IRT-3PL," despite the generalization), where the probability that rater $j$ assigns the correct rating to item $i$ is given by

$$c_n \sim \mathrm{bernoulli}\left(\lambda_i + (1 - \lambda_i) \cdot \mathrm{logit}^{-1}(\delta_i \cdot (\alpha_j^k - \beta_i))\right), \tag{1}$$

where $k = \mathrm{sens}$ if $z_i = 1$ and $k = \mathrm{spec}$ if $z_i = 0$.

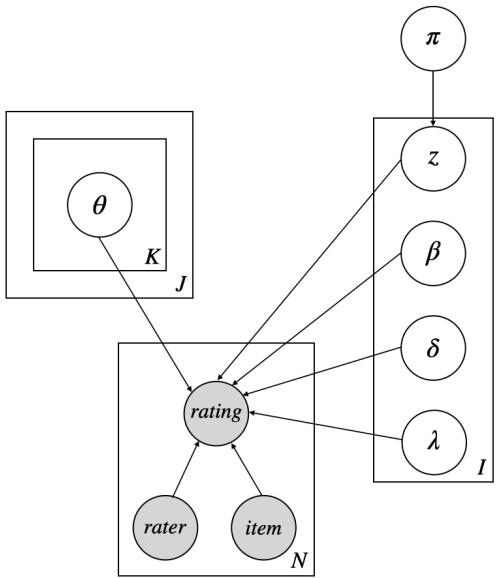

Figure 1: Graphical sketch of the IRT-3PL model. *Sizes*: $J$ number of annotators, $K = 2$ number of categories, $I$ number of items, $N$ number of categories. *Observed data*: labels for $rating, rater$, and $item$. *Parameters*: $\theta$ annotator accuracies/biases, $\pi$ category prevalence, $z$ true item category, $\beta$ item difficulty, $\delta$ item discrimination, $\lambda$ item guessing.

In order to convert to a distribution over ratings, the probability of a 1 outcome must be flipped when $z_i = 0$ so that a 90% accurate rating results in a 90% chance of a 0 rating. Thus the rating is given by

$$y_n \sim \begin{cases} \text{Bernoulli}\left(\lambda_i + (1 - \lambda_i)\cdot \text{logit}^{-1}\left(\delta_i \cdot (\alpha_j^{\text{sens}} - \beta_i)\right)\right), & \text{if } z_i = 1, \\ \text{Bernoulli}\left(1 - \left(\lambda_i + (1 - \lambda_i)\cdot \text{logit}^{-1}\left(\delta_i \cdot (\alpha_j^{\text{spec}} - \beta_i)\right)\right)\right), & \text{if } z_i = 0. \end{cases} \tag{2}$$

The second case ($z_i = 0$) reduces to the expression $\text{bernoulli}\left((1 - \lambda_i) \cdot \left(1 - \text{logit}^{-1}\left(\delta_i \cdot \left(\alpha_j^{\text{spec}} - \beta_i\right)\right)\right)\right)$.

**Spammy and adversarial raters.** A spammy rater is one for whom the rating does not depend on the item being rated, which arises when $\alpha^{\text{sens}} = -\alpha^{\text{spec}}$ (i.e., sensitivity is equal to one minus specificity on the probability scale), so that $\text{logit}^{-1}(\alpha^{\text{sens}}) = 1 - \text{logit}^{-1}(\alpha^{\text{spec}})$ (Passonneau & Carpenter, 2014). For example, a rater with 40% sensitivity and 60% specificity has a 40% chance of returning a 1 rating no matter what the true category is.

An adversarial rater is one for which $\alpha^{\text{sens}} < -\alpha^{\text{spec}}$, which on the probability scale implies sensitivity is less than one minus specificity. We have restricted all models to enforce the constraint that $\alpha^{\text{sens}} > -\alpha^{\text{spec}}$. In contrast, a rater with 60% sensitivity and 60% specificity provides informative ratings, whereas one with 40% sensitivity and 40% specificity provides adversarial ratings which are equally informative despite being more likely to be wrong than correct (Passonneau & Carpenter, 2014).

**Parameter priors** We complete the generalized Bayesian IRT-3PL model by placing weakly informative priors on the parameters. These priors serve to regularize the scale of the parameters and aid identifiability, as recommended by Gabry et al. (2019) and Gelman et al. (2020),

$$\pi \sim \text{Beta}(2, 2), \quad \alpha_j^{\text{spec}} \sim \text{Normal}(2, 2), \quad \alpha_j^{\text{sens}} \sim \text{Normal}(1, 2)$$

$$\beta_i \sim \text{Normal}(0, 1), \quad \delta_i \sim \text{LogNormal}(0, 0.25), \quad \lambda_i \sim \text{Beta}(2, 2).$$

We present the graphical sketch of the IRT-3PL model shown in Figure 1 for clarity, then in Section 4.5 show how this framework reduces to existing crowdsourcing and IRT models by tying or fixing parameters.

### 4.4 Marginal likelihood

A crowdsourcing model generates latent discrete categories $z_i \in \{0, 1\}$ for each item. For both optimization and sampling, it is convenient to marginalize the complete likelihood $p(y, z \mid \pi, \alpha, \beta, \delta, \lambda)$ to the rating likelihood $p(y \mid \pi, \alpha, \beta, \delta, \lambda)$. The marginalization calculation is efficient because it is factored by data item. Letting $\theta = \pi, \alpha, \beta, \delta, \lambda$ be the full set of continuous parameters, the trick is to rearrange the long-form data by item, then marginalize out the discrete parameters, resulting in the likelihood

$$p(y \mid \theta) = \prod_{i=1}^{I} \sum_{z_i=0}^{1} p(z_i \mid \theta) \cdot \prod_{n:\text{item}_n=i} p(y_n \mid z_i, \theta). \tag{3}$$

We start with the complete data likelihood for ratings $y$ and latent categories $z$,

$$p(y, z \mid \theta) = \prod_{i=1}^{I} p(z_i \mid \theta) \cdot \prod_{n=1}^{N} p(y_n \mid z, \theta) \tag{4}$$

and then rearrange terms by item,

$$p(y, z \mid \theta) = \prod_{i=i}^{I} \left( p(z_i \mid \theta) \cdot \prod_{n:\text{item}_n=i} p(y_n \mid z_i, \theta) \right). \tag{5}$$

On a per item basis, the marginalization is tractable, yielding Equation 3. Computational inference requires working on the log scale, where the log marginal likelihood of the rating data in the full model is given by

$$
\begin{aligned}
\log p(y \mid \theta) &= \log \prod_{i=i}^{I} \sum_{z_i=0}^{1} p(z_i \mid \theta) \cdot \prod_{n:\text{item}_n=i} p(y_n \mid z_i, \theta). \\
&= \sum_{i=1}^{I} \text{logSumExp}_{z_i=0}^{1} \left( \log p(z_i \mid \theta) + \sum_{n:\text{item}_n=i} \log p(y_n \mid z_i, \theta) \right),
\end{aligned}
\tag{6}
$$

where

$$\text{logSumExp}_{n=1}^{N} \ell_n = \log \sum_{n=1}^{N} \exp(\ell_n) \tag{7}$$

is the numerically stable log-scale analogue of addition.

### 4.5 Model reductions

By tying or fixing parameters, the full model may be reduced to define a wide range of natural submodels. Six of these models correspond to item-response theory models of the one-, two-, and three-parameter logistic variety, either with or without a sensitivity/specificity distinction. The model with varying rater sensitivity and specificity and no item effects reduces to Dawid and Skene's model. Other models, such as the model with a single item effect and no rater effects have been studied in the epidemiology literature. Table 2 summarizes the possible model reductions and gives them identifying tags.

| Tag | Reduction | Description |
|-----|-----------|-------------|
| A | $\lambda_i = 0$ | no guessing items |
| B | $\delta_i = 1$ | equal discrimination items |
| C | $\beta_i = 0$ | equal difficulty items |
| D | $\alpha^{\text{spec}} = \alpha^{\text{sens}}$ | equal error raters |
| E | $\alpha_i = \alpha_j$ | identical raters |

Table 2: Orthogonal model reductions.

**Tied sensitivity and specificity.** First, we consider models which do not distinguish sensitivity and specificity. Such models should be used when the categories are not intrinsically ordered (e.g., rating two

consumer brands for preference). All of these models other than the last (ABCDE) assumes raters have varying accuracy.

| Reductions | Probability Correct | Note |
|---|---|---|
| $D$ | $\lambda_i + (1 - \lambda_i) \cdot \mathrm{logit}^{-1}(\delta_i \cdot (\alpha_j - \beta_i))$ | IRT 3PL |
| $CD$ | $\lambda_i + (1 - \lambda_i) \cdot \mathrm{logit}^{-1}(\delta_i \cdot \alpha_j)$ | |
| $BD$ | $\lambda_i + (1 - \lambda_i) \cdot \mathrm{logit}^{-1}(\alpha_j - \beta_i)$ | IRT 2PL |
| $BCD$ | $\lambda_i + (1 - \lambda_i) \cdot \mathrm{logit}^{-1}(\alpha_j)$ | |
| $AD$ | $\mathrm{logit}^{-1}(\delta_i \cdot (\alpha_j - \beta_i))$ | |
| $ACD$ | $\mathrm{logit}^{-1}(\delta_i \cdot \alpha_j)$ | |
| $ABD$ | $\mathrm{logit}^{-1}(\alpha_j - \beta_i)$ | IRT 1PL |
| $ABCD$ | $\mathrm{logit}^{-1}(\alpha_j)$ | |
| $ABCDE$ | $\mathrm{logit}^{-1}(\alpha)$ | |

$$(8)$$

**Free sensitivity and specificity.** The following models introduce separate parameters for sensitivity and specificity rather than assuming they are the same. Only the last model (ABCE) does not distinguish rater abilities.

| Reductions | Probability Correct | Note |
|---|---|---|
| | $\lambda_i + (1 - \lambda_i) \cdot \mathrm{logit}^{-1}(\delta_i \cdot (\alpha_j^k - \beta_i))$ | IRT 3PL + sens/spec |
| $C$ | $\lambda_i + (1 - \lambda_i) \cdot \mathrm{logit}^{-1}(\delta_i \cdot \alpha_j^k)$ | |
| $BC$ | $\lambda_i + (1 - \lambda_i) \cdot \mathrm{logit}^{-1}(\alpha_j^k)$ | |
| $A$ | $\mathrm{logit}^{-1}(\delta_i \cdot (\alpha_j^k - \beta_i))$ | IRT 2PL + sens/spec |
| $AC$ | $\mathrm{logit}^{-1}(\delta_i \cdot \alpha_j^k)$ | |
| $AB$ | $\mathrm{logit}^{-1}(\alpha_j^k - \beta_i)$ | IRT 1PL + sens/spec |
| $ABC$ | $\mathrm{logit}^{-1}(\alpha_j^k)$ | Dawid/Skene |
| $ABCE$ | $\mathrm{logit}^{-1}(\alpha^k)$ | |

$$(9)$$

**No rater effects.** The last model, which is common in epidemiology (Albert & Dodd, 2004), includes item effects without any rater effects.

| Reductions | Probability Correct |
|---|---|
| $ABDE$ | $\mathrm{logit}^{-1}(-\beta_i)$ |

$$(10)$$

**Redundant parameter models.** The remaining thirteen models are redundant in the sense that fixing their non-identifiability issues reduces to a model with a single item effect.

| Reductions | Probability Correct |
|---|---|
| $E$ | $\lambda_i + (1 - \lambda_i) \cdot \mathrm{logit}^{-1}(\delta_i \cdot (\alpha^k - \beta_i))$ |
| $DE$ | $\lambda_i + (1 - \lambda_i) \cdot \mathrm{logit}^{-1}(\delta_i \cdot (\alpha - \beta_i))$ |
| $CE$ | $\lambda_i + (1 - \lambda_i) \cdot \mathrm{logit}^{-1}(\delta_i \cdot \alpha^k)$ |
| $CDE$ | $\lambda_i + (1 - \lambda_i) \cdot \mathrm{logit}^{-1}(\delta_i \cdot \alpha)$ |
| $BE$ | $\lambda_i + (1 - \lambda_i) \cdot \mathrm{logit}^{-1}(\alpha^k - \beta_i)$ |
| $BDE$ | $\lambda_i + (1 - \lambda_i) \cdot \mathrm{logit}^{-1}(\alpha - \beta_i)$ |
| $BCE$ | $\lambda_i + (1 - \lambda_i) \cdot \mathrm{logit}^{-1}(\alpha^k)$ |
| $BCDE$ | $\lambda_i + (1 - \lambda_i) \cdot \mathrm{logit}^{-1}(\alpha)$ |
| $AE$ | $\mathrm{logit}^{-1}(\delta_i \cdot (\alpha^k - \beta_i))$ |
| $ADE$ | $\mathrm{logit}^{-1}(\delta_i \cdot (\alpha - \beta_i))$ |
| $ACE$ | $\mathrm{logit}^{-1}(\delta_i \cdot \alpha^k)$ |
| $ACDE$ | $\mathrm{logit}^{-1}(\delta_i \cdot \alpha)$ |
| $ABE$ | $\mathrm{logit}^{-1}(\alpha^k - \beta_i)$ |

$$(11)$$

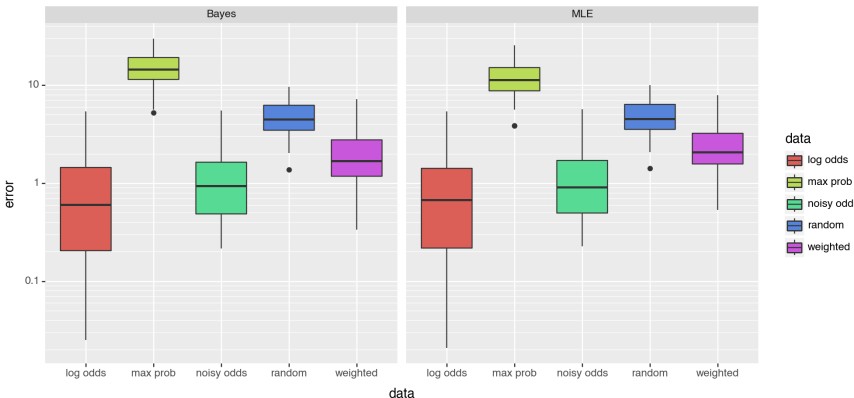

Figure 2: **L2 norm of training error**. The L2 norm of parameter estimation error, $||\widehat{\theta} - \theta||_2$, for different approaches to training with probabilities (lower is better): (log odds) training a linear regression on the log odds, (max prob) assigning the highest probability category, (noisy odds) add standard normal noise to the log odds approach, (random) assign a random category according to the probability, (weighted) train a weighted logistic regression. Estimates and variability are consistent between Bayesian posterior means with a normal prior (left) or ridge-penalized maximum likelihood estimates (right). Regression is 32-dimensional, with correlated inputs and 1024 training data points. Results show standard bar-and-whisker plots over 32 trials with paired random $x, \beta$.

## 5 Training on probabilistic data

As part of their likelihood calculation, the rating models compute the probability that each item in the rating set is either category 0 or category 1. It is far better to use this probabilistic information directly when training a classifier than to collapse the uncertainty to either a 0 or 1 "gold standard" category. Using probabilities acts as a form of regularization—if an item has probability 0.6329, training will try to avoid attenuated results for this item with probabilities near 1 or 0.

Consider a classification data set where for each data item $n$, there is a covariate (feature) vector $x_n \in \mathbb{R}^L$ and a binary outcome $y_n \in \{0, 1\}$. The standard logistic regression likelihood takes a parameter (coefficient, weight) vector $\beta \in \mathbb{R}^L$ with $y_n \sim \text{bernoulli}\left(\text{logit}^{-1}(x_n \cdot \beta)\right)$.

Now suppose that we have fit a rating model and do not actually know $y_n$ but only have an estimate of $\Pr[y_n = 1]$. The conventional strategy in machine-learning data curation is to take the "best" category, often the result of a majority vote. More generally, majority voting is equivalent to taking the highest probability category, $y_i = 1$ if $\Pr[y_i = 1] > \frac{1}{2}$, in a rating model with equal rater accuracies and no item effects (i.e., the model tagged ABCDE).

For the experiment, we generate a synthetic data set of covariates $x_i \sim \text{normal}(0, \Sigma)$, where the positive definite covariance matrix is defined by $\Sigma_{m,n} = \rho^{|m-n|}$, with $\rho = 0.9$. The result is highly correlated covariates; uncorrelated covariates show the same trend. We evaluate five approaches to estimation: (max prob) take $y_n$ to be 1 if $\text{logit}^{-1}(x_n \cdot \beta) > \frac{1}{2}$, (log odds) train a linear regression with outcome $\log x_n \cdot \beta$, (noisy odds) log odds with standard normal noise, $\text{logit}^{-1}(x_n \cdot \beta + \epsilon_n)$, with $\epsilon_n \sim \text{normal}(0, 1)$, (random) randomly generate $y_n \sim \text{bernoulli}\left(\text{logit}^{-1}(x_n \cdot \beta)\right)$ according to its probability distribution, and (weighted) train a weighted logistic regression with outcome 1 and weight $\text{logit}^{-1}(x_n \cdot \beta)$ *and* outcome 0 and weight $1 - \text{logit}^{-1}(x_n \cdot \beta)$.

We provide both Bayesian (posterior mean) and frequentist (penalized maximum likelihood) estimates. For the Bayesian setting, we use a standard normal prior $\beta_k \sim \text{normal}(0, 1)$ and in the frequentist setting we use ridge regression, with penalty function $\frac{1}{2} \cdot \beta^\top \cdot \beta$ to match the Bayesian prior. The resulting error norms in estimating $\beta$ are shown in Figure 2. The plot shows that training with the probabilities is much better than taking the category with the highest probability. The best approach is training a linear regression based on

the log odds, with the noisy version of the same approach not far behind. Weighted training with logistic regression is not quite as good. Randomly selecting a category according to the generative model is not as good as using the weights directly, but it still dominates taking the "best" category.

## 6 Empirical evaluations and ablation studies

The posteriors of all 18 distinct models introduced above were sampled using Markov chain Monte Carlo (MCMC) for two data sets. The first data set consists of 5 dentists rating each of roughly 4000 dental X-rays for caries (a kind of pre-cavity) (Espeland & Handelman, 1989). The second is nearly 200 Mechanical Turkers, each rating a subset of roughly 3000 pairs of sentences for entailment (Snow et al., 2008).

### 6.1 Evaluation metrics

**Posterior predictive checks** We use posterior predictive checks (PPC) for goodness-of-fit testing (Gelman et al., 1996). They are the Bayesian analogue of the $\chi^2$ tests widely employed in epidemiology (Albert & Dodd, 2004), which test a "null" of the fitted model against the observed data (Formann, 2003). PPCs work by generating replicated data from the fitted model and comparing statistics from the replicated data to those from the original data. The posterior predictive distribution for replications $y^{\mathrm{rep}}$ of the original data $y$ given model parameters $\theta$ is $p(y^{\mathrm{rep}} \mid y) = \mathbb{E}\left[p(y^{\mathrm{rep}} \mid \theta) \mid y\right] = \int p(y^{\mathrm{rep}} \mid \theta) \cdot p(\theta \mid y) \, \mathrm{d}\theta$.

If a model fits well, test statistics $s(\cdot)$ should take similar values in both original and replicated data sets. This is summarized by a Bayesian $p$-value-like statistic, $\Pr[s(y^{\mathrm{rep}}) \geq s(y) \mid y] = \int \mathrm{I}(s(y^{\mathrm{rep}}) \geq s(y)) \cdot p(y^{\mathrm{rep}} \mid y) \, \mathrm{d}y^{\mathrm{rep}}$. Any choice of statistic guided by the quantities of interest in the model itself can be used. We focus on marginal positive votes per rater and per item as statistics, because these directly reflect rater-specific biases (sensitivity and specificity) and item-level difficulty. They are simple to interpret, align with classical $\chi^2$ tests, and effectively reject models that omit difficulty when items show variation in difficulty.

**Leave-one-out cross-validation** We use an accurate approximation of leave-one-out cross-validation (LOO) for predictive accuracy (Vehtari et al., 2017). LOO provides a fine-grained view of predictive performance, especially useful for model comparison and refinement.

LOO estimates out-of-sample predictive fit by evaluating the model's performance on each data point, leaving out one observation at a time. This provides an accurate measure of how well the model generalizes to unseen data. The expected log pointwise predictive density ($\mathrm{elpd}_{\mathrm{loo}}$) is computed as $\mathrm{elpd}_{\mathrm{loo}} = \sum_{i=1}^{n} \log p(y_i \mid y_{-i})$, where $p(y_i \mid y_{-i})$ is the predictive density of observation $y_i$, excluding the $i$-th observation. This involves estimating $p(y_i \mid y_{-i}) = \int p(y_i \mid \theta) \, p(\theta \mid y_{-i}) d\theta$, which can be calculated very efficiently using Pareto-smoothed importance sampling given a single model fit (Vehtari et al., 2017).

### 6.2 PPC and LOO performances

The models were coded in Stan (version 2.33) and fit with default sampling settings using CmdStanPy (version 1.20). The default sampler is the multinomial no-U-turn sampler, an adaptive form of Hamiltonian Monte Carlo (Betancourt, 2017; Hoffman et al., 2014) that adapts a diagonal mass matrix. The default number of chains is four, and the default runs 1000 warmup iterations (for burn-in and adaptation) and 1000 sampling iterations. All sampling runs ended with split-$\widehat{R}$ values less than 1.01 for all parameters (prevalence, rater parameters, and item parameters), indicating consistency with convergence to approximate stationarity (Gelman et al., 2013).

Table 3 reports posterior predictive $p$-values and $\mathrm{elpd}_{\mathrm{loo}}$ for all models on the caries and Mechanical Turk natural language inference data. Posterior predictive $p$-values assess how well model-simulated ratings match the observed data, with values near 0.5 indicating good fit. Among models that pass these checks, $\mathrm{elpd}_{\mathrm{loo}}$ provides a measure of predictive accuracy, where higher values indicate better performance. To clarify the structural differences that drive these performance patterns, Table 4 summarizes the key components included in each of the four models.

| Model | Caries | | | NL Inference | | |
|-------|--------|--|--|--------------|--|--|
| | Rater $p$-value | Ratings $p$-value | elpd$_{loo}$ | Rater $p$-value | Ratings $p$-value | elpd$_{loo}$ |
| ABCDE | < 0.001 | < 0.001 | -9604 | < 0.001 | < 0.001 | -5488 |
| ABCD | < 0.001 | < 0.001 | -8797 | < 0.001 | < 0.001 | -5544 |
| ABCE | < 0.001 | 0.019 | -9594 | < 0.001 | 0.011 | -5464 |
| ABDE | < 0.001 | < 0.001 | -9773 | < 0.001 | < 0.001 | -4738 |
| ABC | 0.462 | < 0.001 | -8749 | 0.021 | 0.002 | -5084 |
| ABD | 0.074 | < 0.001 | -8211 | < 0.001 | < 0.001 | -5563 |
| ACD | < 0.001 | < 0.001 | -8669 | < 0.001 | < 0.001 | -5544 |
| BCD | < 0.001 | < 0.001 | -8759 | < 0.001 | < 0.001 | -5546 |
| AB | 0.468 | 0.218 | -8699 | 0.020 | 0.006 | -5096 |
| AC | 0.405 | < 0.001 | -8743 | 0.002 | 0.004 | -5086 |
| AD | 0.325 | < 0.001 | -8192 | < 0.001 | < 0.001 | -5557 |
| BC | 0.102 | 0.001 | -8721 | < 0.001 | < 0.001 | -5127 |
| BD | < 0.001 | < 0.001 | -8446 | < 0.001 | < 0.001 | -5571 |
| CD | < 0.001 | < 0.001 | -8724 | < 0.001 | < 0.001 | -5544 |
| A | 0.046 | 0.289 | -8696 | 0.020 | 0.003 | -5102 |
| C | 0.014 | 0.001 | -8722 | < 0.001 | < 0.001 | -5123 |
| D | < 0.001 | < 0.001 | -8432 | < 0.001 | < 0.001 | -5569 |
| Full | 0.120 | 0.010 | -8678 | < 0.001 | < 0.001 | -5123 |

Table 3: Posterior predictive $p$-values and leave-one-out cross-validation values across Caries and NL Inference (Natural Language Inference) datasets for 18 different models.

| Model | Item Difficulty ($\beta$) | Discrimination ($\delta$) | Guessing ($\lambda$) | Separate Sens/Spec ($\alpha$) |
|-------|--------------------------|--------------------------|---------------------|------------------------------|
| ABC (Dawid–Skene) | ✗ | ✗ | ✗ | ✗ |
| AB (IRT-1PL) | ✓ | ✗ | ✗ | ✓ |
| A (IRT-2PL) | ✓ | ✓ | ✗ | ✓ |
| Full (IRT-3PL) | ✓ | ✓ | ✓ | ✓ |

Table 4: Structured comparison of selected models in terms of item effects and rater parameterization.

For the caries data, only three models pass the posterior predictive checks, all of which are IRT models that include difficulty (AB) and optionally discrimination (A) or guessing (Full). Notably, the widely used Dawid and Skene model (ABC) fails the posterior predictive checks. Among these models, the full model shows the best performance based on elpd$_{loo}$.

For the Mechanical Turk rated entailment data, models A, AB, and ABC pass the posterior predictive checks for raters, with model ABC having the highest elpd$_{loo}$. None of the models look good for ratings $p$-values. The main difference between the Caries and Turker data sets is that there are many more Turkers doing far fewer ratings each, and there are more spammy raters among the Turkers.

## 6.3 Dawid and Skene's posterior predictions underestimate dispersion

To demonstrate how posterior predictions underestimate dispersion, we compare Dawid and Skene's baseline (ABC) with three item-level models: difficulty (AB), plus discrimination (A), and plus guessing (Full) in Amazon Turker data. The goodness-of-fit tests are similar, so in Figure 3, we plot the expected number of marginal positive votes for models A, AB, ABC, and Full along with the actual data. The Dawid and Skene model (ABC) has an inflated number of middling positive votes compared to the data; in contrast the full model underestimates them. However, posterior predictive inference for models A (ability plus difficulty plus discrimination) and AB (ability plus difficulty) more closely matches the actual data. This alignment enhances the model's ability to reflect true voter consensus, thereby producing a distribution of votes that

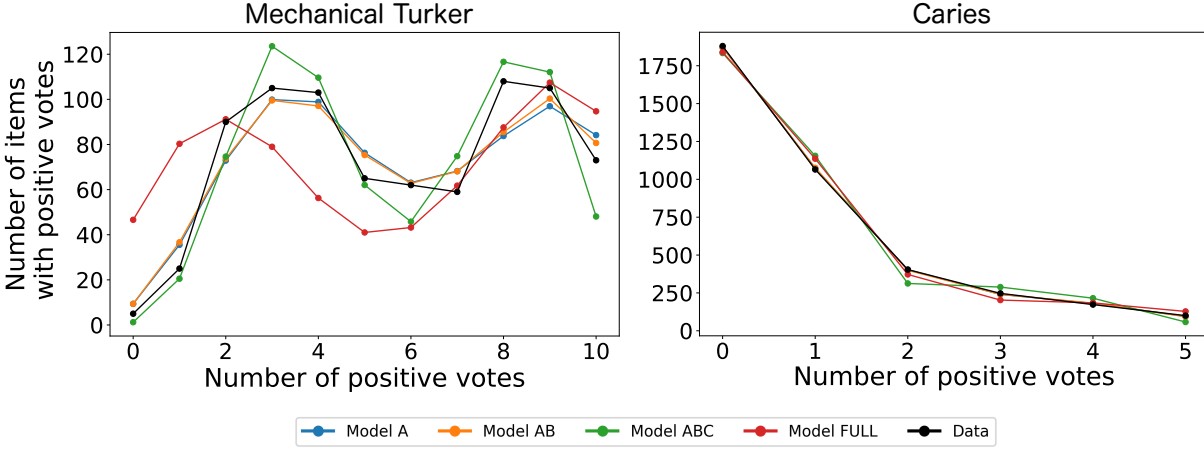

Figure 3: Distribution of positive votes per item comparing the baseline Dawid & Skene model (ABC) with IRT models with difficulty (AB), discrimination (A), and guessing (Full) alongside actual data, demonstrating the varying levels of dispersion captured by each model.

more accurately mirrors observed voting patterns. For the caries data, the story is similar. Models A, AB, and Full closely match the actual data, while the Dawid and Skene model (ABC) again inflates the number of middling positive votes compared to the real data.

# 7 Extensions

**Including covariate information**   Parameterizing on the log odds scale makes it straightforward to add features (i.e., covariates) in addition to the baseline random effects for items or raters. The simple models presented here can be thought of as intercept-only versions of more general models.

Features for raters might include demographic information such as education, location, native language, age, education level, whether they are an AI and which one, etc. Features for items can be used to inform difficulty, discrimination and guessability. For example, a covariate might indicate the number of options in a multiple choice test, the length of sentences used for inference, or the grade level of the textbook from which the item was culled, when a person's last dental checkup was, etc.

If item-level or rater-level covariates are available, they may be used to inform the parameters in the usual way through a regression in the form of a generalized linear model (Gelman & Hill, 2007). For example, suppose there are item-level covariates $x_i \in \mathbb{R}^K$. With a parameter vector $\gamma \in \mathbb{R}^K$, the generative model for the category of an item may be extended to a logistic regression, $z_i \sim \text{bernoulli}\left(x_i^\top \cdot \gamma\right)$.

**$K$-way categorical rating**   A natural extension is to $K$-way categorical ratings, such as classifying a dog image by species, classifying an article in a newspaper by topic, rating a movie on a one to five scale, classifying a doctor's visit with an ICD-10 code, and so on. Most of the work on ratings has been in this more general categorical setting. With more than two categories, sensitivity and specificity are replaced with categorical responses based on the latent true category. Discrimination and guessing act the same way, but difficulty must be replaced with a more general notion of a categorical item level effect, which may represent either focused alternatives (e.g., a border collie is confusable with an Irish shepherd) or diffuse (e.g., can't tell what's in the image).

**Population-level models**   With enough raters, these models may also be extended hierarchically to make population-level inferences about the distribution of rater abilities or item difficulties (Paun et al., 2018). Several of the crowdsourcing tasks may be combined to select raters and items to rate online with active

learning, which is a form of reinforcement learning. With a hierarchical model, inference may be expanded to new raters (Paun et al., 2018).

**Ordered, count, and other data**   It is also straightforward to extend a rating model to ordered responses such as Likert scales (Lakshminarayanan & Teh, 2013; Rogers et al., 2010; Shatkay et al., 2005), rank ordering (Chen et al., 2013; Rafailov et al., 2024), counts (the "textbook" case of crowdsourcing is estimating the number of jelly beans in a container (Surowiecki, 2005)), proportions/probabilities, distances, or pairs of real numbers such as planetary locations (Smyth et al., 1994). All that needs to change is the response model and the representation of the latent truth—the idea of getting noisy ratings and inferring a ground truth remains. As an example, Smyth had raters mark images of Venus for volcano locations (Smyth et al., 1994). The true location is represented as a latitude and longitude and rater responses can be multivariate normal centered around the true, but unknown, location. For ordinal ratings, an ordinal logit model of the truth may be used (Rogers et al., 2010). For comparisons, the Bradley-Terry model can be used (Bradley & Terry, 1952), and for ranking, the Plackett-Luce generalization (Plackett, 1975; Luce, 1959), as used in direct preference optimization for fine-tuning large language models (Rafailov et al., 2024).

**Joint estimation of a classifier**   When item-level covariates are available, Raykar et al. (2010) provide an approach to jointly estimating the parameters of the rating model and the classifier. In essence, the prevalence model is updated to a logistic regression classifier and the the classifier participates jointly in rating along with the raters.

## 8   Conclusion

Until this work, models used for crowdsourcing categorical responses, such as that of Dawid & Skene (1979), produced biased results. We demonstrated this bias through failed goodness-of-fit and LOO results. Dawid and Skene's model predicts too little inter-annotator agreement relative to the actual ratings observed in the wild. By introducing item-level effects, the models presented here are able to adjust for item difficulty, discrimination, and guessability, resulting in better fits. The enhanced fits not only pass the goodness-of-fit tests for both items and ratings, but also show improved performance based on LOO, providing a more robust evaluation of predictive accuracy.

Dawid and Skene's model has two modes, an adversarial and cooperative one. Adversarial solutions arise when the majority of raters have below-chance accuracy. We showed how to parameterize the models to ensure a cooperative solution. In crowdsourcing, it is more common to find random (aka "spammy") annotators than adversarial ones; constraining the models to a cooperative solution identifies the model, but should not be used if the data contains a mix of adversarial and cooperative raters (Passonneau & Carpenter, 2014). Dealing with majority adversarial raters is an open problem.

Models of annotation introduce significant improvements compared to the conventional approach of evaluating inter-annotator agreement statistics such as Cohen's $\kappa$; Paun et al. (2022) provide an overview. They may also be used to diagnose rater biases, infer adjusted population prevalence, or select highly discriminative items, and so on. We also showed how using uncertainty of the form generated by crowdsourcing models, either as weights or to directly train or to sample, improves classifier training.

The bottom line is that most training efforts in machine learning are not making the most of their human feedback and can be improved by applying the methods introduced here.

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
