# OpenReview forum: "Crowdsourcing with Difficulty: A Bayesian Rating Model for Heterogeneous Items"
_TMLR — Rejected by TMLR_

### Review · Reviewer_Kuk6 · 2025-07-03

**Summary Of Contributions:**

This work proposes a Bayesian rating model for binary ratting data that incorporates item-level effects for difficulty, discriminativeness, and guessability, as well as rater-specific sensitivity and specificity. The authors also demonstrate how to assess model fit using posterior predictive checks and evaluate predictive accuracy through leave-one-out cross-validation.

**Audience:**

Yes

**Broader Impact Concerns:**

I do not consider this work having any ethical concerns.

**Claims And Evidence:**

Yes

**Requested Changes:**

Please see in the weakness.

**Strengths And Weaknesses:**

Stength:

- This is a highly flexible model that enables inference on a wide range of item-level effects, as outlined in my summary, and significantly extends earlier crowdsourcing models such as Dawid & Skene (1979). I especially appreciate the discussion of reduced models in Section 4.4.
- The writing is generally clear and effectively motivates the introduction of item-level parameters. I found the paragraph explaining the five tasks to which crowdsourcing models are applied particularly informative.
- I briefly reviewed the code; it is well-documented and appears easy to use.

Weakness:

- For pedagogical purposes and to improve readability, I believe it would be helpful to introduce models from prior work in Section 4.2. It is not immediately clear what the novel contributions of the model in this section are. While I understand that the authors briefly discuss this in the model reduction section, a more explicit comparison earlier on would aid comprehension.

- The choice of summary statistics for the posterior predictive checks (PPC) was not discussed comprehensively. Are the marginal positive votes per rater and per item the default choices of statistics for rater models? What other summary statistics could be considered in this context?

- Despite the flexibility of the model, it does appear to involve a large number of parameters. Section 4.4 also discusses several possible model reductions. What potential issues might arise from using an overly complex model? Could this lead to non-identifiability or poorly behaved posterior distributions? And how can users perform efficient model selection aside from doing PCC for all the submodels?

---

### Review · Reviewer_QSEZ · 2025-08-03

**Summary Of Contributions:**

This paper proposed a crowdsourcing model that taking the item-level effect into account. The model is able to adjust based on item level effects like difficaulty. Experiments on real-world datasets showed improvements on fitting.

**Audience:**

Yes

**Broader Impact Concerns:**

No issue from my perspective

**Claims And Evidence:**

No

**Requested Changes:**

1. The whole structure of the paper need to be changed. By reading Section 1, reader without expertise in crowdsourcing will have no knowledge about what is crowdsoucing model. I suggest re-organized the structure of section 1 as follows.

    New Section 1: A quick introduction about crowdsourcing -> a summarized introduction about the applications -> Introduce rating data -> define crowdsourcing problem and crowdsourcing model (and the role of model in problem) -> what question this paper is trying to solve -> The main contribution of this paper (please introduce Figure 1, or move it to where it is used) -> previous works.

    The application of crowdsourcing can be either section 2 or send to appendix.

2. In section 4.2 it's not clear how the priors' distributions are determined? Are the prarameter learned from data or something else?

3. I suggest re-writing section 4.1 - 4.3, I used more than 4 hours on this section, but not able to understand what the proposed model is. Note have background in crowdsourcing.

**Strengths And Weaknesses:**

Strength: I'm anot able to analysis because the presentation needs a huge improvement.

Weakness: The presentation needs a huge improvement (see the requested changes below)

---

> ### Author Response · Authors · 2025-08-29
> **Response to feedback**
>
> **1.** Thank you for this helpful suggestion. We have reorganized the introduction so that it now begins with a concise overview of crowdsourcing, followed by a brief summary of applications, an introduction to rating data, the definition of the crowdsourcing problem and model, the central research question, and finally the main contributions. To further streamline the flow, the discussion of applications has been moved to Section 3, after previous work. In addition, we relocated the material on posterior predictive checks and leave-one-out cross-validation from the introduction to Section 6, where the empirical evaluations are presented. Please refer to the revised PDF.
>
> **2.** These priors are not learned from the data; they are fixed, weakly informative distributions chosen to regularize the model by constraining parameters to plausible scales and to improve identifiability. Please refer to the revised PDF.
>
>
> **3.** Thank you for pointing this out. In the revised version, we have reorganized Section 4 into a clearer flow. Section 4.1 now introduces the rating data and notation; Section 4.2 presents prior crowdsourcing models; and Sections 4.3–4.5 describe our proposed model, including the generative process, marginal likelihood, and model reductions. In addition, we relocated Figure 1 into Section 4 so that it is introduced consistently alongside the proposed model.

---

### Review · Reviewer_VLfQ · 2025-08-16

**Summary Of Contributions:**

This paper introduces a Bayesian crowdsourcing model that extends Dawid and Skene’s classic approach by incorporating item-level effects: difficulty, discrimination, and guessability. Traditional models only adjust for annotator sensitivity and specificity, often underestimating variability and biasing downstream training. The authors show that their model better captures the distribution of rating data, avoids adversarial solutions by constraining parameters, and provides improved inference of ground truth.

They validate the approach with two datasets—dental X-rays rated for cavities and natural language inference annotations—demonstrating superior model fit via posterior predictive checks and leave-one-out cross-validation. Results show that training with probabilistic labels outperforms majority voting or deterministic selection.

The paper contributes: (1) a generalized item-response-based rating model, (2) a method to handle adversarial or spammy raters, and (3) open-source implementations. Overall, it argues that most machine learning training pipelines underutilize human feedback, and adopting these methods can significantly improve reliability and accuracy.

**Audience:**

Yes

**Broader Impact Concerns:**

Not applicable.

**Claims And Evidence:**

Yes

**Requested Changes:**

1. A structured comparison—for example, a table that contrasts the proposed method against related models along dimensions such as treatment of item difficulty, discrimination, guessing, or rater sensitivity—would provide much-needed clarity.

2. Placement and Referencing of Figures. Figure 1 is introduced at the very beginning of the paper (top of page 2), but the first substantive discussion of it does not occur until page 6. This mismatch between figure placement and textual reference forces the reader to flip back and forth unnecessarily, reducing readability and flow. Figures should be introduced closer to where they are discussed.

3. Typos and Editorial Issues.
Several typos and grammatical issues reduce the professionalism of the manuscript:
* “whihc” → “which”
* “presentted” → “presented”
* “enhanced fits not only not only pass” → remove the repeated phrase.
Careful proofreading is needed to eliminate these errors and improve readability.

**Strengths And Weaknesses:**

I could not infer any significant strength, so I will just list some weaknesses.

# Clarity of the Introduction
The introduction is overly lengthy and difficult to parse. The central scientific question is buried beneath extended background information, making it hard for readers to quickly grasp what problem the paper addresses and why it matters. The five bolded subsections on pages 2–3 appear disconnected, and their relationships are not clearly articulated. This creates unnecessary cognitive load for the reader. Furthermore, deferring the contributions list until page 4 diminishes the impact; readers benefit from seeing the key contributions early on to contextualize the subsequent discussion.

# Organization of Section 4
Section 4 interweaves background material on existing crowdsourcing models with the authors’ proposed innovations, making it difficult to disentangle prior work from novel contributions. This blending obscures both the motivation for the new model and the originality of the paper.

# Inconsistencies in Formatting and Interpretation
The use of bolded numbers is inconsistent and unexplained. Without a clear rule, this formatting distracts rather than guides the reader. In Table 3, the interpretation of evaluation metrics is also confusing. It appears that both p-values and expected log pointwise predictive density (elpd) are treated as performance indicators, but their directionality (i.e., whether higher or lower is better) is not explicitly explained. For example, the Full model does not consistently outperform alternatives on both datasets, yet it is described as the strongest. This raises doubts about whether the evaluation criteria are being applied and interpreted correctly.

---

> ### Author Response · Authors · 2025-08-29
> **Response to feedback**
>
> **Clarity of the Introduction**
>
> Thank you for this helpful suggestion. We have reorganized the introduction so that it now begins with a concise overview of crowdsourcing, followed by a brief summary of applications, an introduction to rating data, the definition of the crowdsourcing problem and model, the central research question, and finally the main contributions. To further streamline the flow, the discussion of applications has been moved to Section 3, after previous work. In addition, we relocated the material on posterior predictive checks and leave-one-out cross-validation from the introduction to Section 6, where the empirical evaluations are presented. Please refer to the revised PDF.
>
> **Organization of Section 4**
>
> Sure, that’s a good point. In the revised version, we have reorganized Section 4 into a clearer flow: Section 4.1 introduces the rating data and notation; Section 4.2 is prior crowdsourcing models; and Sections 4.3–4.5 present our proposed model, its generative process, marginal likelihood, and model reductions. In addition, we relocated Figure 1 into Section 4 so that it is introduced consistently alongside the proposed model.
>
> **Inconsistencies in formatting and interpretation**
>
> Thank you for the suggestion. We removed bolding from Table 3 to avoid confusion and added explicit directionality: Posterior predictive p-values assess how well model-simulated ratings match the observed data, with values near 0.5 indicating good fit. Among models that pass these checks, elpd_loo provides a measure of predictive accuracy, where higher values indicate better performance. The goal of Table 3 is not to claim a single model dominates, but to illustrate how model components affect fit across datasets. For caries, only three IRT-based models with item difficulty (AB, A, Full) pass the PPC, with Full best on elpd_loo; Dawid–Skene (ABC) fails despite being widely used. For mechanical turk, A, AB, and ABC pass the PPC for raters, with ABC best on elpd_loo, though none fit well under the ratings p-value due to spammy raters. These results highlight both trade-offs in model design and differences in data quality, as the mechanical turk setting involves noisier, more inconsistent raters.
>
> **A structured comparison**
>
> We thank the reviewer for this suggestion. We have added Table 4 to clarify how our proposed model relates to and extends existing approaches.
>
> **Placement and Referencing of Figures & Typos**
>
> We thank the reviewer for pointing out these issues. Figure 1 is now introduced and discussed in Section 4, directly where the model is explained. We also corrected the noted typos and carefully proofread the manuscript to improve overall readability.

---

### Decision · Action_Editor_ZSXD · 2025-10-02

**Recommendation:** Reject

**Additional Comments:**

Reviewers expressed major concerns regarding the presentation, including an overly lengthy and difficult-to-parse introduction (Section 1) and the interweaving of background material on existing crowdsourcing models with the proposed methods (Section 4), which made it difficult to disentangle prior work from the paper’s contributions. As a result, two reviewers were unable to clearly identify the strengths of the manuscript. The third reviewer also noted that improvements in readability are required.

Even after revisions, reviewers found the manuscript difficult to follow, with contributions insufficiently distinguished from background material and the presentation requiring considerable effort to read. Moreover, the experimental results were viewed as inconsistent in supporting the central claims, with performance varying substantially across datasets. On this basis, two reviewers recommended rejection and one recommended acceptance.

**Audience:**

Yes

**Audience Explanation:**

Yes.

**Claims And Evidence:**

No

**Claims Explanation:**

The central claim is that by introducing item-level effects, the proposed models can adjust for item difficulty, discrimination, and guessability, thereby achieving better model fits. These improved fits not only satisfy goodness-of-fit tests for both items and ratings but also demonstrate enhanced predictive accuracy as evidenced by leave-one-out (LOO) evaluations. The first dataset comprises ratings from five dentists on approximately 4,000 dental X-rays for caries (a type of pre-cavity) (Espeland & Handelman, 1989). The second dataset involves nearly 200 Mechanical Turkers, each rating a subset of roughly 3,000 sentence pairs for entailment (Snow et al., 2008). Results vary across the two datasets, therefore, the claims made in the submission are not supported by accurate, convincing, or sufficiently clear evidence.